# Zinc oxide nanoparticles modulate the gene expression of $ZnT_1$ and $ZIP_8$ to manipulate zinc homeostasis and stress-induced cytotoxicity in human neuroblastoma SH-SY5Y cells

Chien-Yuan Pan[1], Fang-Yu Lin[2], Lung-Sen Kao[3,4], Chien-Chang Huang[4], Pei-Shan Liu[2]*

1 Department of Life Science and Institute of Zoology, National Taiwan University, Taipei, Taiwan,
2 Department of Microbiology, Soochow University, Taipei, Taiwan, 3 Brain Research Center, National Yang-Ming University, Taipei, Taiwan, 4 Department of Life Sciences and Institute of Genome Sciences, National Yang-Ming University, Taipei, Taiwan

* psliu@scu.edu.tw

**Data Availability Statement:** All relevant data are within the paper and its Supporting Information files.

## Abstract

Zinc ions ($Zn^{2+}$) are important messenger molecules involved in various physiological functions. To maintain the homeostasis of cytosolic $Zn^{2+}$ concentration ($[Zn^{2+}]_c$), Zrt/Irt-related proteins (ZIPs) and $Zn^{2+}$ transporters (ZnTs) are the two families of proteins responsible for decreasing and increasing the $[Zn^{2+}]_c$, respectively, by fluxing $Zn^{2+}$ across the membranes of the cell and intracellular compartments in opposite directions. Most studies focus on the cytotoxicity incurred by a high concentration of $[Zn^{2+}]_c$ and less investigate the $[Zn^{2+}]_c$ at physiological levels. Zinc oxide-nanoparticle (ZnO-NP) is blood brain barrier-permeable and elevates the $[Zn^{2+}]_c$ to different levels according to the concentrations of ZnO-NP applied. In this study, we mildly elevated the $[Zn^{2+}]_c$ by ZnO-NP at concentrations below 1 μg/ml, which had little cytotoxicity, in cultured human neuroblastoma SH-SY5Y cells and characterized the importance of $Zn^{2+}$ transporters in 6-hydroxy dopamine (6-OHDA)-induced cell death. The results show that ZnO-NP at low concentrations elevated the $[Zn^{2+}]_c$ transiently in 6 hr, then declined gradually to a basal level in 24 hr. Knocking down the expression levels of $ZnT_1$ (located mostly at the plasma membrane) and $ZIP_8$ (present in endosomes and lysosomes) increased and decreased the ZnO-NP-induced elevation of $[Zn^{2+}]_c$, respectively. ZnO-NP treatment reduced the basal levels of reactive oxygen species and *Bax/Bcl-2* mRNA ratios; in addition, ZnO-NP decreased the 6-OHDA-induced ROS production, *p53* expression, and cell death. These results show that ZnO-NP-induced mild elevation in $[Zn^{2+}]_c$ activates beneficial effects in reducing the 6-OHDA-induced cytotoxic effects. Therefore, brain-delivery of ZnO-NP can be regarded as a potential therapy for neurodegenerative diseases.

**Funding:** This work was supported by grants from Ministry of Science and Technology, Taiwan, R. O. C. (https://www.most.gov.tw/?l=en) NSC 102-2320-B-031-001, MOST 104-2622-B-031-001-CC2, & MOST 104-2632-B-031-001 to PSL and MOST 107-2320-B-002-052 to CYP. The funder had no role in study design, data collection and analysis, decision to publish, or preparation of the manuscript.

**Competing interests:** The authors have declared that no competing interests exist.

## Introduction

Zinc ion ($Zn^{2+}$) is essential for all living organisms and is the second most abundant transition element in human. It is a cofactor in many proteins regulating their catalytic activities and structure. In addition, recent emerging evidence has shown that $Zn^{2+}$ is a messenger in regulation of many cellular activities such as cell cycle, cell proliferation, differentiation and death via different signaling pathways [1, 2]. Cytosolic $Zn^{2+}$ concentration ($[Zn^{2+}]_c$) changes during cell cycle, differentiation and cell death [3]. During cell proliferation, the tyrosine phosphatases are suppressed by a small elevation of $[Zn^{2+}]_c$ to activate ERK pathway [4]. A number of transcription factors, such as p53, contain $Zn^{2+}$ binding motifs affecting cell cycle and survival [5].

The paradoxical, but vital, roles of $Zn^{2+}$ in nervous system have gained recognition recently [6, 7]. $Zn^{2+}$ is essential for neurogenesis, neuronal differentiation and synaptic transmission. The inhibition of synaptic $Zn^{2+}$ signaling in hippocampus and amygdala by $Zn^{2+}$ chelators affects cognition [8]. $Zn^{2+}$ deficiency reduces neurogenesis and associates with neuronal dysfunction. A correlation between $Zn^{2+}$ deficiency and depression has been demonstrated in both clinical studies and animal models [9, 10]. In contrast, high $Zn^{2+}$ levels block mitochondrial function and induce apoptosis in the development of pathophysiology of CNS disorders including epilepsy, schizophrenia and Alzheimer's Disease [11]. At cellular level, high dose of $Zn^{2+}$ is neurotoxic causing cell death [12–14] and $Zn^{2+}$ deficiency causes caspase-dependent apoptosis in human neuronal precursor cells [15, 16]. $Zn^{2+}$ supplementation significantly reduces spinal cord ischemia-reperfusion injury in rats [17]. However, dietary $Zn^{2+}$ supplementation has restrictions and limitations in crossing brain-blood barrier (BBB), which has limited permeability for $Zn^{2+}$, especially when the desired final $Zn^{2+}$ level is higher than physiological levels [18]. Thus, controlled and targeted delivery of $Zn^{2+}$ is highly desirable.

Nanoparticles (NP) technologies have been used for the targeted delivery of chemicals [19]. In nervous system, polylactide-co-glycolide or BBB ligand specific-modified polylactide polymers are used to carry $Zn^{2+}$ across BBB [18, 19]. However, the rate is slow, the cellular or brain entrance are evidenced after several days [19]. We have previously demonstrated the entrance of zinc oxide-NP (ZnO-NP) into brain via olfactory bulb in rat and elevates the $[Zn^{2+}]_c$ in cultured cells [20]. Therefore, ZnO-NP has the potential to be a potent means for $Zn^{2+}$ delivery to regulate $[Zn^{2+}]_c$ homeostasis in the central nervous system.

The cellular uptake of ZnO-NP into intracellular compartments is via endocytosis followed by dissolution that occurs in acidic compartments to convert ZnO-NP to $Zn^{2+}$ [20]. Two classes of proteins are implicated in $Zn^{2+}$ transport for $[Zn^{2+}]_c$ homeostasis: solute-linked carrier 30 (SLC30, Zn transporter (ZnT)) and SLC39 (Zrt/Irt-realted proteins (ZIP)) decrease and increase the $[Zn^{2+}]_c$, respectively, by fluxing $Zn^{2+}$ across the membranes of cell and intracellular organelles in opposite directions. The ZIP proteins then transport the accumulated $Zn^{2+}$ in these acidic compartments to the cytosol and ZnT proteins work corporately to flux $Zn^{2+}$ out of the cytosol. Therefore, ZnO-NP may be different from direct $Zn^{2+}$ application in regulating expression levels of $Zn^{2+}$ transporters to control $Zn^{2+}$ homeostasis.

ZnO-NP at high dosage causes apoptosis in lung [21] and neural stem cells [13] and interferes with the ion channel activities in primary cultured rat hippocampal neurons [22]. However, toxicity is not seen under exposure to ZnO-NP at low doses, such as 6 ppm (70 μM) [13], or 10 μM [20]. The importance of $Zn^{2+}$ to normal functioning of the central nervous system is increasingly appreciated [9, 15]. In this report, we mildly elevated the $[Zn^{2+}]_c$ in human neuroblastoma cells, SH-SY5Y, by ZnO-NP at concentrations below 1 μg/ml. ZnO-NP treatment greatly enhanced the expression level of $ZnT_1$ and less affected the expression of $ZIP_8$. ZnO-NP treatment decreased the basal level of reactive oxygen species (ROS) and the expression ratio of Bax/Bcl-2. In addition, ZnO-NP treatment recued the cell death caused by the

6-hydroxy dopamine (6-OHDA). Therefore, BBB-permeable ZnO-NP provides a therapeutic strategy to treat neurodegeneration disorders by fin-tuning the $[Zn^{2+}]_c$.

## Materials and methods

### Chemicals

ZnO-NPs were purchased from Sigma-Aldrich Co. (St. Louis, MO, USA). Their preparation protocols were described in our previous work [21]. The size range of ZnO-NP in solution was from 20 to 80 nm with an average of 45 nm. SH-SY5Y neuroblastoma cells were purchased from the American Type Culture Centre CRL2266 (Manassas, VA, USA). FluoZin-3-AM, Lipofectamine 2000®, reverse transcriptase III and TRIzol® reagent were purchased from Invitrogen Co. (Carlsbad, CA, USA). RNase-free DNAse I and RNeasy purification columns were purchased from Qiagen Inc. (Valencia, CA, USA). Random hexamer primers were obtained from Fermentas Inc. (Burlington, Canada). iQ SYBR Green Supermix was obtained from Bio-Rad Inc. (Hercules, CA, USA). Other chemicals were obtained from Merck KGaA (Darmstadt, Germany) otherwise indicated.

### Cell culture

Human neuroblastoma SH-SY5Y cells were cultured in minimal essential medium (Gibico 41500–034) supplemented with F12 nutrient mixture (Gibico 21700–075) and 10% fetal bovine serum. The cells were kept in a humidified 5%-$CO_2$ incubator at 37 ºC [20].

### $[Zn^{2+}]_c$ Measurements

Suspended cells were incubated in a Loading buffer (in mM, NaCl 150, glucose 5, Hepes 10, $MgCl_2$ 1, KCl 5, $CaCl_2$ 2.2, pH7.3) containing 10 μM of FluoZin-3-AM at 37˚C for 30 minutes. After washing out the FluoZin-3-AM by centrifugation and resuspending the cell in Loading buffer, the changes in the fluorescence intensity were recorded as described before [20].

### RT-PCR assay

RNA extraction and reverse transcription were performed following the protocols suggested by the manufactures. The primers for the polymerase chain reactions (PCR, Q-Amp™ 2x Hot-Start PCR Master Mix) were listed in S1 Table in S1 File. The products were separated by electrophoresis on 2% agarose gels, stained with ethidium bromide, and photographed with ultraviolet trans-illumination. For quantitative PCR (qPCR), the kit used was $IQ^2$ Fast qPCR System and the instrument was from Illumina Inc. (Eco™ Real-time PCR system) [23].

### ROS measurements

To quantify the production of ROS, we loaded the cells with 2',7'-dichlorodihydrofluorescein diacetate ($H_2$DCFDA, Molecular Probes®) and incubated at 37℃, 5% $CO_2$ for 30 minutes. After replacing the medium, 6-OHDA or $H_2O_2$ were added. The fluorescence intensities were measured by a microplate reader (Glomax-multidetection system, Promega, USA) with excitation at 485 nm and emission at 500–560 nm.

### $ZIP_8$ and $ZnT_1$ shRNA knockdown

Plasmids expressing short hairpin RNAs (shRNA) against $ZIP_8$ and $ZnT_1$ were purchased from National RNAi Core Facility, Academia Sinica, Taiwan, and the target sequences of these shRNAs (4 for $ZIP_8$ and 5 for $ZnT_1$) were listed in S2 Table in S1 File. Lipofectamine 2000®

was used to transfect these plasmids into SH-SY5Y cells [24]. An apoplasmid was used as negative control.

## MTT assay

The MTT assay, an index of cell viability and cell growth, is based on the ability of viable cells to reduce MTT (3-(4,5-dimethylthiazol-2-yl)-2,5-diphenyl tetrazolium bromide) [25]. All samples were assayed in triplicate from 5 batches of cells. The total time of ZnO-NP treatment was 24 hr and 6-OHDA was added at $18^{th}$ hr. To enhance cell death caused by 6-OHDA, cells were incubated in a medium containing 0.5% of serum. For cells after shRNA transfection, the serum was 5% during all experiments.

## Statistical analysis

Statistical analysis was performed using one-way analysis of variance and significant differences were assessed by Student's *t* test. A *p* value less than 0.05 was regarded as statistically significant.

## Results

### ZnO-NP elevates $[Zn^{2+}]_c$ in cultured SH-SY5Y cells

To examine ZnO-NP at low doses can elevate $[Zn^{2+}]_c$ in cultured human neuroblastoma SH-SY5Y cells, we loaded the cells with FluoZin3, a $Zn^{2+}$-sensitive dye, and monitored the changes in fluorescence intensities (Fig 1). The addition of ZnO-NP (0.081 and 0.814 μg/ml, n = 3 each) increased the fluorescence intensity gradually during the 200-s recording period in a concentration-dependent manner. For 25-hr long-term treatment, the fluorescence intensities measured reached a maximum in 6 hr when treated with different concentrations of ZnO-NP (0.081, 0.814, and 8.14 μg/ml, n = 3 each and statistical symbols were shown in S1 Fig in S1 File). These results reveal that ZnO-NP apparently elevates the $[Zn^{2+}]_c$ transiently in a concentration- and time-dependent mode even at low concentrations.

### $ZnT_1$ and $ZIP_8$ regulate the ZnO-NP-induced $[Zn^{2+}]$ responses in SH-SY5Y cells

ZIPs and ZnTs play important roles in maintaining the $[Zn^{2+}]_c$ homeostasis. We first characterized the expression levels of *ZnT* and *ZIP* isoforms in cultured SH-SY5Y cells by RT-PCR and the results showed significant expressions of $ZnT_1$, $ZnT_3$, $ZnT_4$, $ZnT_5$, $ZnT_6$, $ZnT_7$, $ZnT_9$ and $ZnT_{10}$ (S2A Fig in S1 File) and $ZIP_1$, $ZIP_3$, $ZIP_4$, $ZIP_6$, $ZIP_7$, $ZIP_8$, $ZIP_9$, $ZIP_{10}$, $ZIP_{11}$, $ZIP_{13}$ and $ZIP_{14}$ (S2B Fig in S1 File). $ZnT_1$ is the main transporter at the plasma membrane to efflux $Zn^{2+}$ out of cells and lowers the $[Zn^{2+}]_c$ [26]; $ZIP_8$ presents in the synaptic vesicles and lysosomes to transport $Zn^{2+}$ from intracellular compartments to the cytosol [27, 28]. Since endocytosis is the main route for ZnO-NP entrance into the cell and dissolution into $Zn^{2+}$ occurs in an acidic compartment [20], we focused on characterizing the involvement of $ZnT_1$ and $ZIP_8$ in modulating the ZnO-NP-induced $[Zn^{2+}]_c$ response in SH-SY5Y cells (Fig 2). We adopted qPCR to investigate the mRNA levels of $ZnT_1$ and $ZIP_8$ in SH-SY5Y cells after the addition of ZnO-NP of different concentrations (Fig 2A & 2B, n = 3 for each concentration). The average results show that a low-dose of ZnO-NP (0.081 μg/ml) elevated the expression levels of $ZnT_1$ and $ZIP_8$ transiently in 6 hr (*p* < 0.05) and then declined to a basal level after 24 hrs. High doses of ZnO-NP (0.814 and 8.14 μg/ ml) treatment maintained the expression of $ZnT_1$ at a level 4~8 fold higher than the control group (*p* < 0.05) during the 24-hour exposure period. ZnO-NP at 0.814 μg/ml elevated and maintained the expression of $ZIP_8$ at a level 2–3 fold

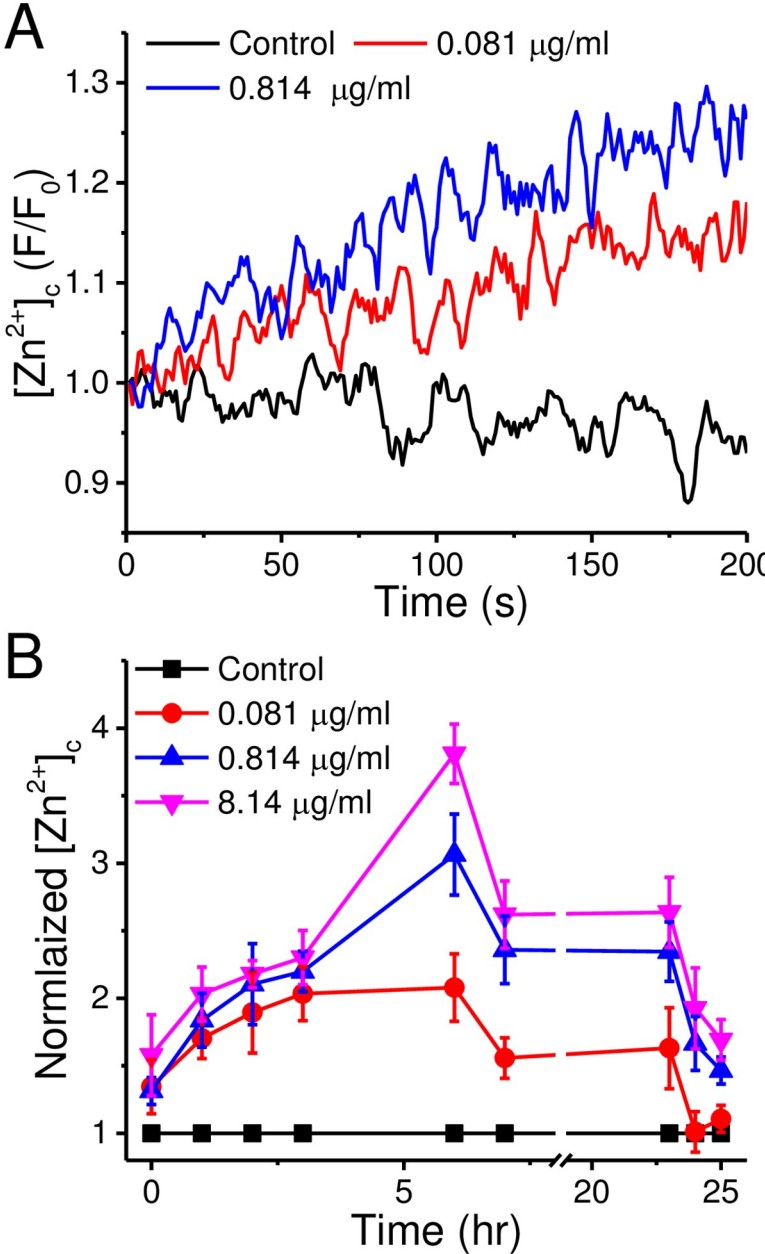

**Fig 1. ZnO-NP exposure induces a transient elevation of $[Zn^{2+}]_c$ in SH-SY5Y cells.** We loaded the cells with FluoZin-3 and monitored the changes of the fluorescence intensities from a group of suspended cells stimulated with different concentrations of ZnO-NPs. A. The short-term $[Zn^{2+}]_c$ responses. ZnO-NP (0, 0.081, and 0.814 µg/ml) were added at the beginning of the recording and the fluorescence intensities were normalized to the value at the time zero $(F/F_0)$. B. Normalized $[Zn^{2+}]_c$ elevation after long-term exposure of ZnO-NP. The fluorescence intensities from ZnO-NP-treated suspension cells were normalized to the control group without ZnO-NP treatment (Normalized $[Zn^{2+}]_c$) at different time after ZnO-NP exposure. Data presented were Mean ± SEM from 3 batches of cells; *, **, and *** indicates the Student's $t$-test $p$ value $< 0.05$, 0.01, and 0.001, respectively.

higher than the control group ($p < 0.05$), however, at 8.14 µg/ml, ZnO-NP had little effect on the expression of $ZIP_8$. These results reveal that ZnO-NP exposure differentially enhances the expression of $ZnT_1$ and $ZIP_8$.

To verify the contributions of these transporters in regulating the $[Zn^{2+}]_c$ responses induced by ZnO-NP, we delivered specific shRNAs into the cells to reduce the translation of

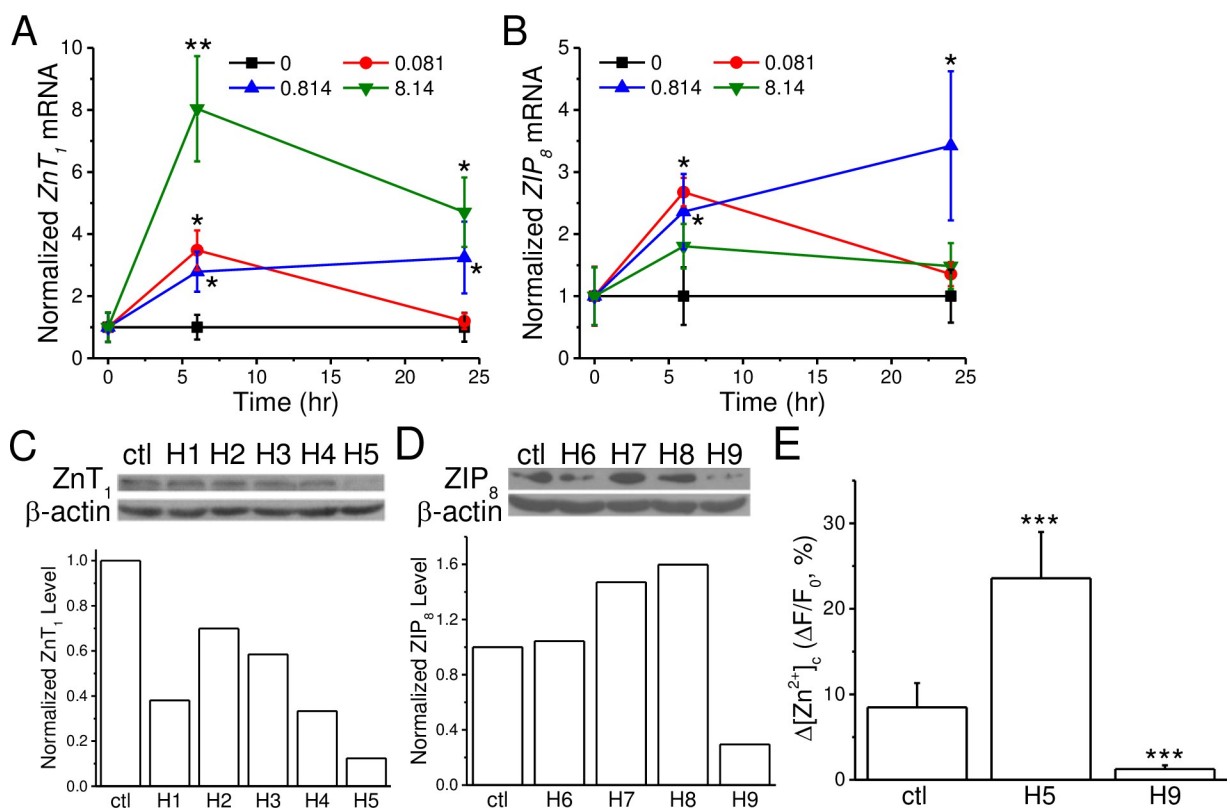

**Fig 2. Knockdown the expressions of specific Zn²⁺ transporters interfere [Zn²⁺]$_c$ responses in SH-SY5Y cells.** A. and B. The expression levels of *ZnT1* and *ZIP8*, respectively. Cells were treated with different concentrations of ZnO-NP for 0, 6 and 24 hr and the mRNA levels of *ZnT₁* and *ZIP₈* were analyzed by RT-PCR. The expression levels were normalized to that of β-actin. C. and D. Expression knockdown of *ZnT1* and *ZIP8*, respectively. Specific shRNAs against *ZnT₁* (H1-5) and *ZIP₈* (H6-9) were delivered into the cells for 1 day and the protein levels were examined by Western blot (upper panel). The intensities of each protein bands were normalized to that of β-actin (lower panel). E. [Zn²⁺]$_c$ responses in transfected cells. Cells were transfected with H5 and H9 shRNAs for 1 day and then loaded with FluoZin3. The changes in the fluorescence intensities (ΔF/F$_0$) induced by ZnO-NP (0.814 μg/ml) were calculated. Data presented were Mean ± S.E.M from 3 bathes of cells. *, **, ***: $p < 0.05$, 0.01, and 0.001, respectively (Student's *t*-test) when compared to the control group.

*ZnT₁* and *ZIP₈* (Fig 2C & 2D, respectively) (The original images of the Western blot were shown in S3 Fig in S1 File). The results of the Western blots revealed that most of these shRNAs decreased the protein levels of *ZnT₁* (H1-5) and *ZIP₈* (H6-9); among them, H5 and H9 were the most effective shRNAs in reducing the protein levels of *ZnT1*, by 88%, and *ZIP₈*, by 70%, respectively. Treating transfected SH-SY5Y cells with ZnO-NP (0.814 μg/ml, Fig 2E, n = 3), the averaged changes in [Zn²⁺]$_c$, comparing to the control group, was about 4-fold higher in cells expressing H5 ($p < 0.001$) and mostly abolished in cells expressing H9 ($p < 0.001$). It is likely that cells change the expression levels of these transporters to regulate the [Zn²⁺]$_c$ in response to different stimulations.

## ZnO-NP at a low dose increases the *Bax/Bcl-2* expression level

To characterize the toxicity of ZnO-NP on SH-SY5Y cells, we treated the cells with different concentrations of ZnO-NP for 24 hr and monitored the viability by MTT assay (Fig 3A). The results show that ZnO-NP exposure reduced the viability in a dose dependent manner with an EC$_{50}$ of 6.8 ± 0.2 μg/ml (n = 15). Under 2 μg/ml, ZnO-NP had little effect on cell viability. We then examined the expression levels of *Bax* and *Bcl-2* by qPCR in SH-SY5Y cells treated with

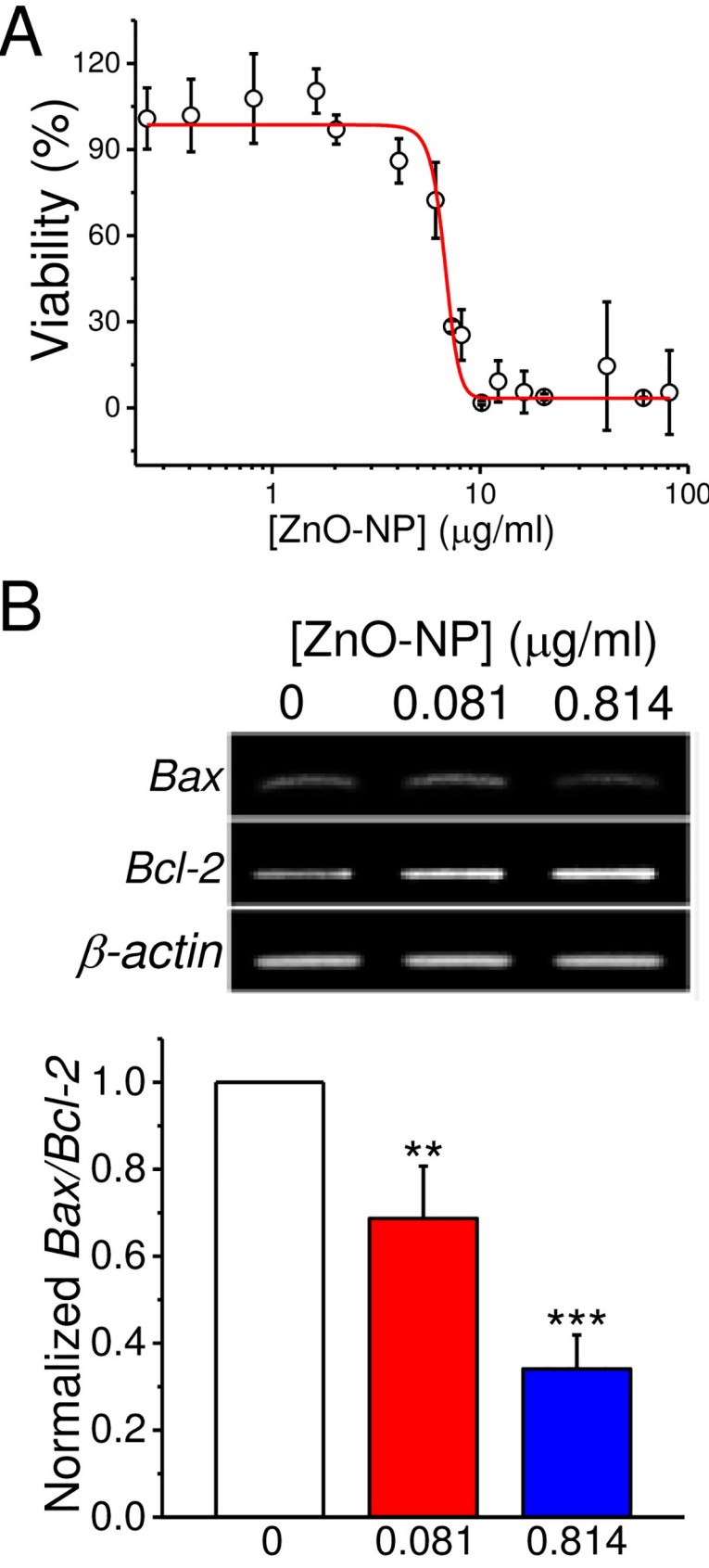

**Fig 3. Low-dose ZnO-NP exposure reduces basal apoptosis signal in SH-SY5Y cells.** A. Dose-dependent cell viability. After a 24-hr ZnO-NP exposure at different concentrations, the cell viability was analyzed by an MTT assay. The dose-dependence were fitted by a Boltzmann equation with an $EC_{50}$ of $6.8 \pm 0.2$ μg/ml. Data presented were Mean ± SEM from 5 batches of cells. B. The *Bax/Bcl-2* ratio. Cells were treated with ZnO-NP for 24 hr and then the mRNA were collected for RT-PCR to analyze the expression levels (upper panel) of *Bax* and *Bcl-2*. The intensities of the PCR products were normalized to the level of *β-actin* and then used to calculate the *Bax/Bcl-2* ratio (Lower panel). Data presented were Mean ± S.E.M (n = 3). ** and ***: $p < 0.01$ and 0.001, respectively, by Student's *t*-test when compared to the control group.

ZnO-NP at 0.081 and 0.814 μg/ml for 24 hr (Fig 3B). The amounts of the PCR products expressed from *Bax* and *Bcl-2* decreased and increased, respectively, as the concentrations of ZnO-NP increased (the original images of the agarose gel analysis of the PCR products were shown in the S4 Fig in S1 File). After normalization, the *Bax/Bcl-2* expression ratio were significantly decreased to $0.69 \pm 0.12$ ($p < 0.01$) and $0.34 \pm 0.08$ ($p < 0.001$), respectively. In contrast, ZnO-NP at 8.14 μg/ml significantly increased the ratio to $1.49 \pm 0.2$ (n = 3, $p < 0.01$, not shown). Therefore, that ZnO-NP at low non-lethal doses decreases the *Bax/Bcl-2* ratio indicating the blockage of apoptosis pathway.

ROS accumulation can trigger the expression of apoptosis-related genes. We then examined the intracellular ROS levels by loading the cells with $H_2DCFDA$ and monitored the changes in the fluorescence intensities in 2 hr (S5 Fig in S1 File). For control cells without ZnO-NP treatment, the ROS level increased over the recording period; in the presence of ZnO-NP (0.081 and 0.814 μg/ml), the ROS levels at the same duration were lower than that of the control group. These findings suggest that a low-dose exposure of ZnO-NP elicits beneficial effects in cells to reduce the oxidation stress and protect cells from death.

## ZnO-NP counteracts stress-induced ROS generation and cell death in SH-SY5Y cells

The uptake of 6-OHDA, an analog of dopamine, into cells through dopamine transporters triggers the production of ROS and causes cell death. To verify ZnO-NP has a protective effect on the 6-OHDA-induced cell death, we pretreated the SH-SY5Y cells with a low dose of ZnO-NP (0.081 and 0.814 μg/ml) for 18 hr then 6-OHDA of different concentrations were added for another 6 hr (Fig 4A, n = 5). The results show that 6-OHDA at 50 and 100 μM significantly decreased the cell viability from $99.9 \pm 7.0\%$ of control group to $67.3 \pm 10.3$ ($p < 0.01$) and $42.1 \pm 4.3$ ($p < 0.01$), respectively. ZnO-NP pretreatment counteracted the 6-OHDA-induced cell death but not significant when 6-OHDA applied was 50 μM; in contrast, for 6-OHDA at 100 μM, the viabilities were significantly enhanced to $64.2 \pm 12.9$ ($p < 0.01$) and $53.4 \pm 12.7\%$ ($p < 0.05$) by ZnO-NP at 0.081 and 0.814 μg/ml, respectively. In addition, ZnO-NP (0.081 μg/ml) pretreatment significantly reduced the ROS level to $80 \pm 6\%$ of the Control group (n = 15, $p < 0.05$) and 6-OHDA treatment greatly elevated the ROS level to $135 \pm 10\%$ (n = 15, $p < 0.05$) (Fig 4B). ZnO-NP pretreatment could significantly reduce this increment to $98 \pm 11\%$ (n = 15, $p < 0.05$) (Fig 4B). Furthermore, ZnO-NP pretreatment reversed the effects of $H_2O_2$ in cell survival and ROS production (S6 Fig in S1 File). We then used RT-PCR to examine the expression level of *p53*, a transcription factor involved in the ROS-activated apoptosis pathway [29], in SH-SY5Y cells (Fig 4C and the original images were shown in S7 Fig in S1 File). After normalization to that of the control group, 6-OHDA treatment significantly increased the expression level of *p53* to $1.5 \pm 0.3$ (n = 3, $p < 0.01$) and this increment could be reduced by the pretreatment of ZnO-NP (0.081 μg/ml) to $0.9 \pm 0.0$ (n = 3, $p < 0.05$). These results suggest that ZnO-NP at a concentration below 1 μg/ml suppresses the production of ROS and reduced the expression of *p53* to facilitate cell survival.

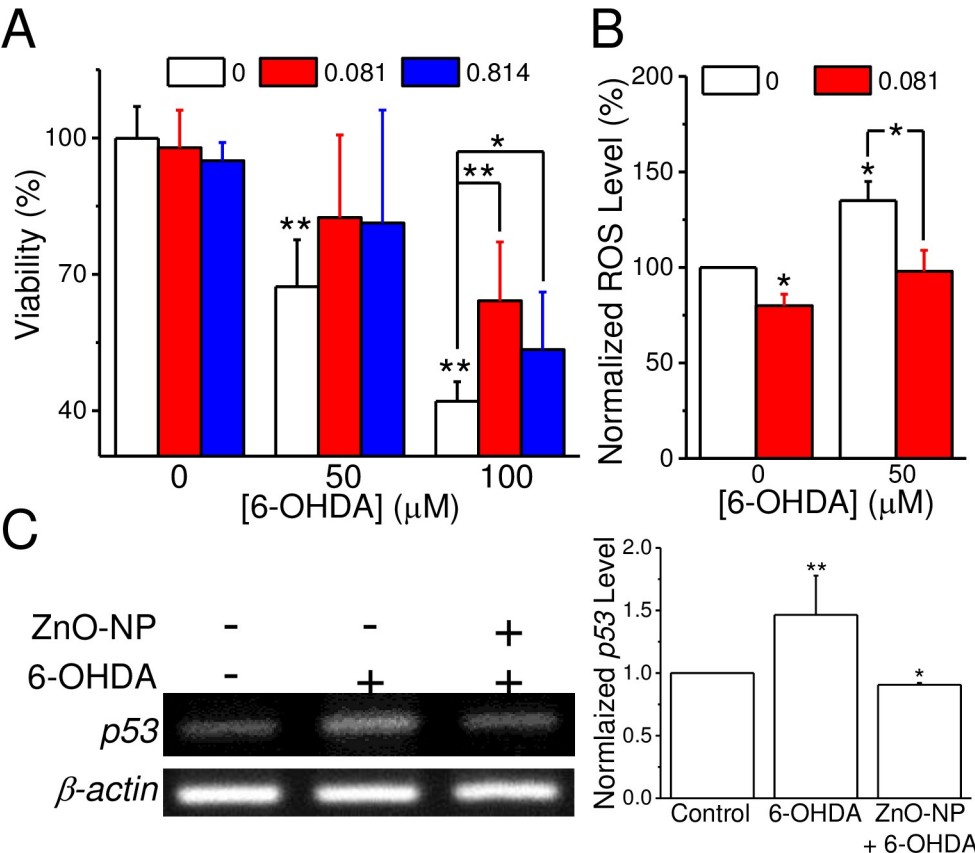

**Fig 4. ZnO-NP suppresses 6-OHDA-induced cytotoxicity in SH-SY5Y cells.** Cells were incubated in a medium containing ZnO-NP (0, 0.081, 0.814 μg/ml) for 24 hr and 6-OHDA (50 or 100 μM) was added at 18th hr. A. Cell viability. The viability was measured by an MTT assay from 5 batches of cells. B. Normalized ROS production. Data presented were Mean ± SEM (n = 15). C. *p53* mRNA levels. The ZnO-NP used was 0.081 μg/ml and cells were harvested for RT-PCR. The density of *p53* products were normalized to that of β-actin and Control group (n = 3). The significance was analyzed by Student's *t*-test; *, **: $p < 0.05$ and 0.01, respectively, when compared to the control group without 6-OHDA treatment or as indicated.

## ZnT1 and ZIP8 knockdown affected 6-OHDA-induced cytotoxicity

To verify the importance of ZnO-NP-induced elevation of $[Zn^{2+}]_c$ in protecting cells from death, we transfected the SH-SY5Y with shRNAs against *ZnT1* and *ZIP8*, then examined the cell viability under 6-OHDA treatment with MTT assay (Fig 5, n = 15). Because of the damages caused by the transfection reagents, the culture medium contained 5% of serum during the experiment. The results show that 6-OHDA (50 μM) treatment decreased the viability from 100.8 ± 4.7 to 87.6 ± 5.1% ($p < 0.01$). Knockdown the expression of *ZnT1* recused the cell death caused by 6-OHDA to a level similar to that of the control group and the addition of ZnO-NP did not enhance the viability. In contrast, *ZIP8* knockdown did not have such a protective effect in 6-OHDA-induced cell death (100.0 ± 1.9 vs. 94.1± 3.0%) and the addition of ZnO-NP did not reverse the toxic effect of 6-OHDA. As shown in Fig 2E, knockdown the expression of *ZnT1* and *ZIP8* enhanced and suppressed the ZnO-NP-induced elevations of $[Zn^{2+}]_c$, respectively. Therefore, the release of $Zn^{2+}$ from the acidic compartments by ZIP8 and the elevation of $[Zn^{2+}]_c$ facilitated by ZnT1 are important in enhancing the viability of cells under different challenges.

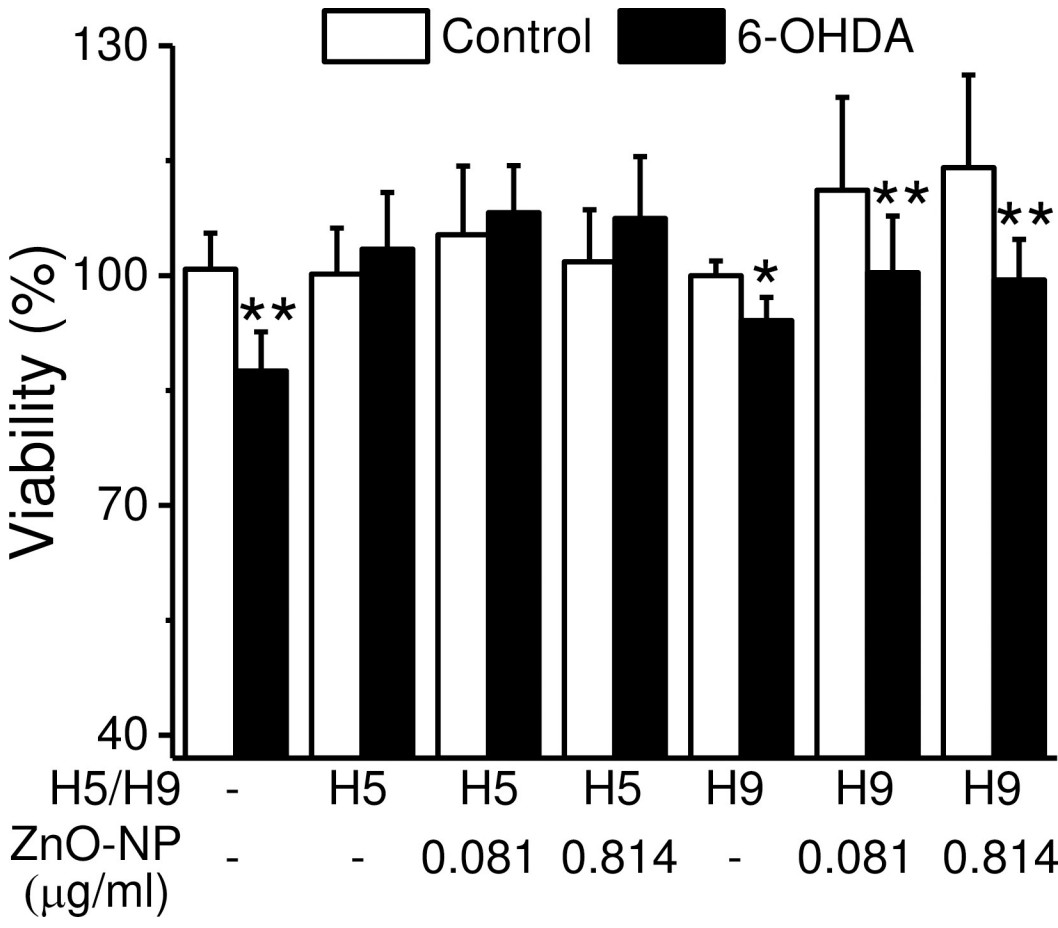

**Fig 5. ZnO-NP altered 6-OHDA-induced cytotoxicity in cells with transporter knockdown.** H5 and H9 shRNAs were transfected into SH-SY5Y cells for 24 hr to knock down the expression of $ZnT_1$ and $ZIP_8$, respectively. Cells were then treated with ZnO-NP of different concentrations for 18 hr and then 6-OHDA (50 μM) was added for another 6 hr. The cell viability was determined by MTT assay. Data presented were Mean ± SEM (n = 5 batches) and the significance were analyzed by Student's $t$-test; * and **: $p < 0.05$ and 0.01, respectively, when compared to the group without 6-OHDA treatment.

## Discussion

This study finds that ZnO-NP potently induced the expressions of $ZnT_1$ and $ZIP_8$ to modulate $[Zn^{2+}]_c$, a crucial parameter for cytoviability in human neuroblastoma SH-SY5Y cells. Below lethal dosage under 1 μg/ml, ZnO-NP transiently elevated the $[Zn^{2+}]_c$ and decreased the *Bax/Bcl-2* expression ratio. In addition, ZnO-NP suppressed the cytotoxicity, ROS production and *p53* gene expression induced by 6-OHDA or $H_2O_2$. These results suggest the cell-protective function of ZnO-NP at low dosages against oxidative stresses and support a therapeutic strategy by delivering ZnO-NP into the CNS to suppress the development of neuropathological disorders.

$Zn^{2+}$ trafficking was investigated in these experiments. ZnO-NP-induced changes of $[Zn^{2+}]_c$ were studied in cells transfected with shRNA against $ZnT_1$ to illustrate the role of $ZnT_1$ for the efflux of $Zn^{2+}$. $[Zn^{2+}]_c$ and the expression of $ZnT_1$ were coupled; both showed increases under exposure to low doses of ZnO-NP and returned to the basal levels after 24 hr. At high dosage (8.14 μg/ml), ZnO-NP induced a large increase in $[Zn^{2+}]_c$, coupled with an 8-fold increase in $ZnT_1$ mRNA (at 6 hr). In this case, both the expression level of $ZnT_1$ and $[Zn^{2+}]_c$

remained high throughout the observation period. Moreover, neurotoxicity induced by 6-OHDA was suppressed in the *ZnT1*-kockdowned cells. Our data show that $[Zn^{2+}]_c$ changes are coupled with the *ZnT_1* expression levels which are closely related to the neuron-protection activity of $Zn^{2+}$. $ZnT_1$ is known to be a plasma membrane protein that is enriched in postsynaptic dendritic spines and plays a role in $Zn^{2+}$ homeostasis in synaptic neuron functions and diseases [30]. Su *et al*. reported a positive correlation between $ZnT_1$ and $Zn^{2+}$ content in the spinal cord [31], and $ZnT_1$ is shown to increase significantly with progression of Alzheimer's disease [32].

Our data suggest that changes in *ZnT_1* expression can become a marker for $[Zn^{2+}]_c$ disturbance associated with neuroviability. Other ZnTs such as $ZnT_{10}$, at Golgi, is down-regulated by an elevation of extracellular $Zn^{2+}$ in SH-SY5Y cells [33]. IL-6 induces a down-regulation of *ZnT_{10}* and enhances the accumulation of $Mn^{2+}$ that might be correlated with Parkinson's disease [34]. Further studies on ZnTs, ZIPs, and metallothioneins (MTs), are required to understand their roles in modulating the $Zn^{2+}$ homeostasis.

We have previously demonstrated the internalization of ZnO-NP by PC12 cells upon exposure to the ZnO-NPs for 10 min. Furthermore, after nasal exposure to airborne ZnO-NP, the NPs are found in rat brain under a transmission electron microscope [20]. We also verify that ZnO-NP elevates $[Zn^{2+}]_c$ in both cultured cells and rat white blood cells through endocytosis and subsequent dissolution in acidic compartments such as endosomes [21]. Conversion of ZnO to ions following entrance into lysosomes has also been shown in the studies of Xia *et al*. in which the labeled ZnO is traced in BEAS-2B cells [35]. Muller *et al*. also have demonstrated that ZnO dissolves rapidly in a lysosomal fluid at a pH of 5.2 [36].

$ZIP_8$ has been shown to be localized in the lysosomal membrane and synaptosomes [27, 28]. Our data show that ZnO-NP-induced $[Zn^{2+}]_c$ changes are greatly suppressed in *ZIP_8*-knockdowned cells, illustrating that *ZIP_8* is required for intracellular $Zn^{2+}$ release from those organelles after ZnO-NP was engulfed, which may be the main route for ZnO in elevating $[Zn^{2+}]_c$. The mRNA levels of *ZIP_8* and *ZnT_1* were positively correlated with the changes in $[Zn^{2+}]_c$ under exposure to ZnO-NP below 1 µg/ml. At a high dose of ZnO-NP (8.14 µg/ml), the expression of $ZIP_8$ was small in contrast to $[Zn^{2+}]_c$ response and *ZnT_1* expression. The low level of $ZIP_8$ prevent additional $Zn^{2+}$ fluxing to the cytosol and further cellular damage. These results suggest that there is a negative feedback between elevation of $[Zn^{2+}]_c$ and the expression of *ZIP_8*.

ROS is known to cause DNA damage that activates the *p53*-linked apoptosis pathway through phosphorylation by ATM. *Bcl-2* has been shown to be coupled with the pro-survival pathway to counteract the effects of mitochondrial damages induced by *Bax*. In addition, silencing the expression of *ZnT_1*, but not *ZIP_8*, can not only enhance the ZnO-NP-induced $[Zn^{2+}]_c$ elevation but rescue the 6-OHDA-induced cell death. It is likely that $[Zn^{2+}]_c$ response is a perquisite for ZnO-NP to reduce stress-induced cytotoxicity by suppressing ROS generation and augmenting expression of *bcl-2*.

$Zn^{2+}$ has been widely shown as a potential antioxidant for suppression of apoptosis [37–44]. In animal brain studies, $Zn^{2+}$ treatment decreases the *Bax/Bcl-2* protein ratio [44]; treating SH-SY5Y cells with a low dose of $Zn^{2+}$ can reverse a stress-induced increment of DNA fragmentation [12]. $Zn^{2+}$ supplementation can reduce the levels of ROS to prevent cardiomyocyte apoptosis and congenital heart defects [40]; it also promotes the recovery of spinal cord function [17, 45]. $Zn^{2+}$ has a protective effect on renal ischemia-reperfusion injury by augmenting superoxide dismutase activity and lowering the *Bax/Bcl-2* expression ratio to reduce apoptosis [37]. Therefore, our results support that ZnO-NP, at sub-lethal dosage, causes a mild elevation of $[Zn^{2+}]_i$ which has been shown to enhance the expression of metallothioneins [46] and the activity of $Zn^{2+}$-related superoxide dismutase [47]. Both metallothioneins and dismutase can

lower the ROS level leading to the down-regulation of the expression of *p53* and *Bax/Bcl-2* ratio [47, 48]. In the future, we will further characterize how $[Zn^{2+}]_i$, when elevated to different levels, regulates the ROS production.

In this and previous studies, we show that ZnO-NP dose-dependently exert paradoxical protective and cytotoxic functions through their ability to alter $[Zn^{2+}]_c$ and modulate the expression of *ZnT₁* and *ZIP₈*. Delivering ZnO-NP at a low dose into the central nervous system may provide a practical strategy to elevate the $[Zn^{2+}]_c$ for potent neuroprotection. Further studies, both *in vivo* and *in vitro*, will be required using more sensitive and selective techniques to measure the homeostasis of $[Zn^{2+}]_c$ and to assess the feasibility of using ZnO-NP for clinical application.

## Supporting information

**S1 File.**
(DOCX)

## Acknowledgments

We wish to thank Ms. Suzanne Hosier for English editing and Hui-Hsing Hung for the $[Zn^{2+}]_c$ measurements in cultured neuronal cells.

## Author Contributions

**Data curation:** Fang-Yu Lin.

**Funding acquisition:** Chien-Yuan Pan, Pei-Shan Liu.

**Methodology:** Lung-Sen Kao, Chien-Chang Huang.

**Supervision:** Chien-Yuan Pan, Pei-Shan Liu.

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
