## [Decision Letter · Decision Letter 0]

11 Jun 2020

PONE-D-20-11050

Zinc oxide nanoparticles modulate the gene expression of ZnT1 and ZIP8 to manipulate zinc homeostasis and stress-induced cytotoxicity in human neuroblastoma SH-SY5Y cells

PLOS ONE

Dear Dr. Liu,

Thank you for submitting your manuscript to PLOS ONE. After careful consideration, we feel that it has merit but does not fully meet PLOS ONE’s publication criteria as it currently stands. Therefore, we invite you to submit a revised version of the manuscript that addresses the points raised during the review process.

ACADEMIC EDITOR:

The manuscript #PONE-D-20-11050 entitled "Zinc oxide nanoparticles modulate the gene expression of ZnT1 and ZIP8 to manipulate zinc homeostasis and stress-induced cytotoxicity in human neuroblastoma SH-SY5Y cells" has been reviewed for publication in PLoS ONE. The reviewers’ and additional editor comments are copied below.

We look forward to receiving your revised manuscript.

Kind regards,

Academic Editor

PLOS ONE

Journal Requirements:

3. Please provide additional information about each of the cell lines used in this work, including source, history and any quality control testing procedures (authentication, characterisation, and mycoplasma testing). For more information, please see http://journals.plos.org/plosone/s/submission-guidelines#loc-cell-lines.

4. To comply with PLOS ONE submission guidelines, in your Methods section, please provide additional information regarding your statistical analyses. For more information on PLOS ONE's expectations for statistical reporting, please see https://journals.plos.org/plosone/s/submission-guidelines.#loc-statistical-reporting.

Additional Editor Comments (if provided):

1. Please add the number of experiments for each statistical result.

2. In Fig. 4A, the inhibitory effect of ZnO-NP in 6-OHDA-induced cell death does not appear to be in a dose-dependent manner. Please check this.

3. In Fig. 2S, why does the ROS of the control group increase with time? Does the author do any stimulation to this group? Please explain it.

4. In Fig. 4B, I think that ZnO-NP seems to only inhibit basal ROS, but does not affect the 6-OHDA-stimulated ROS production. Please check and explain it.

In addition, please mark the unit of ROS production in the figure.

5. In Fig. 5, 6-OHDA (50 �M) reduced cell viability (~15%), which is inconsistent with the data of Fig. 4A (~40%). This may cause the results of Fig. 5 to be misjudged. Please check and redo it.

6. Please add the statistical symbol in the figures and clearly explain in the legends, including in Fig. 1, 2A, 2B, and 3A. This will increase their credibility.

Reviewers' comments:

Reviewer's Responses to Questions

**Comments to the Author**

1. Is the manuscript technically sound, and do the data support the conclusions?

Reviewer #1: Yes

Reviewer #2: Yes

2. Has the statistical analysis been performed appropriately and rigorously? 

Reviewer #1: Yes

Reviewer #2: Yes

3. Have the authors made all data underlying the findings in their manuscript fully available?

Reviewer #1: Yes

Reviewer #2: Yes

4. Is the manuscript presented in an intelligible fashion and written in standard English?

Reviewer #1: Yes

Reviewer #2: Yes

5. Review Comments to the Author

Reviewer #1: Generally speaking, this paper is clear and concise. The maintext is perfect and well

written. However, there are several points that should be carefully revised.

1. In figure 4, 5, S2 and S3, the incubation times of ZnO-NP treated cells were difference. The authors should explain why the difference.

2. The authors need to discuss the possible reason or mechanism for explaining why ZnO-NP decreased ROS production and p53 expression.

3. Please indicate the molecular size of ZnT1 and ZIP8 in figure 2. The exact migration position of at least one independent molecular weight marker protein should be indicated with each gel or blot. Each blot panel should retain space above and below the band of interest from the original image.

4. Some spelling errors should be avoided. Such as, in discussion 9th line, “ snRNA” change “shRNA”.

Reviewer #2: Authors investigated the effects of Zinc oxide nanoparticles, which had little cytotoxicity, in cultured human neuroblastoma SH-SY5Y cells and characterized the importance of Zn2+ transporters in 6-hydroxy dopamine (6-OHDA)-induced cell death. The manuscript is well-written, with a significant summary and discussion.

6. PLOS authors have the option to publish the peer review history of their article (what does this mean?). If published, this will include your full peer review and any attached files.

Reviewer #1: No

Reviewer #2: No

---

## [Author Response · Author response to Decision Letter 0]

13 Aug 2020

Questions from academic editor:

Q1. Please add the number of experiments for each statistical result.

Response: Thanks for the suggestion. We have included the number of experiments and statistical results in the main text and figure legends. 

Q2. In Fig. 4A, the inhibitory effect of ZnO-NP in 6-OHDA-induced cell death does not appear to be in a dose-dependent manner. Please check this.

Response: The large variations of the results shown in Fig. 4A make it was difficult to evaluate that ZnO-NP suppressed the cell death in a dose-dependent manner. However, Fig. 5 and Fig. S5 show that ZnO-NP at these concentrations had a similar effect in suppressing the cell death and reducing the production of ROS, respectively. Henceforth, it is likely that the protecting effects of ZnO-NP at these concentrations are at the plateau phase against the 6-OHDA-evoked oxidative stresses. 

Q3. In Fig. 2S, why does the ROS of the control group increase with time? Does the author do any stimulation to this group? Please explain it.

Response: Fig S2 is now Fig S5. There is no treatment other than the ZnO-NP application in this experiment. The increase may come from the cellular normal metabolism which produces ROS continuously as suggested by the manufacture. Some reports consider this increment as the background and subtract them from the results. 

Q4. In Fig. 4B, I think that ZnO-NP seems to only inhibit basal ROS, but does not affect the 6-OHDA-stimulated ROS production. Please check and explain it. In addition, please mark the unit of ROS production in the figure.

Response: Thanks for the suggestion. We reorganized the results by normalizing the responses to that of the control group in each independent batch of cells. The average results showed that ZnO-NP treatment reduced the ROS level to 80 +/- 6% (p < 0.05) and 6-OHDA treatment significantly elevated the ROS level to 135 +/- 10% (p < 0.05). This increment in ROS production was significantly reduced to 98 +/- 11% by ZnO-NP pretreatment (p < 0.05) (n = 15 for each group). Therefore, these results support that a low dose of ZnO-NP enhances the cell survival by reducing the generation of ROS. (Page 10, Lines 216-219) 

Q5. In Fig. 5, 6-OHDA (50 uM) reduced cell viability (~15%), which is inconsistent with the data of Fig. 4A (~40%). This may cause the results of Fig. 5 to be misjudged. Please check and redo it.

Response: Sorry for the confusion. The cell survival experiment of Fig. 4A was performed in a medium containing 0.5% of serum in order to reveal the toxicity of 6-OHDA. However, SH-SY5Y cells became fragile after transfection (Fig. 5) and could not survive well in a medium containing 0.5% of serum for 6-OHDA treatment. Thus the experiment shown in Fig. 5 was performed in a medium containing 5% of serum during 6-OHDA treatment. We have modified the manuscript to reveal the difference. (Page 6 Line 130-134; Page 10 Lines 232-234)

Q6. Please add the statistical symbol in the figures and clearly explain in the legends, including in Fig. 1, 2A, 2B, and 3A. This will increase their credibility.

Response: Thanks for the suggestion. Fig. 1B is a line graph shows the changes in the [Zn2+]c during the incubation of ZnO-NP at different concentrations, it was difficult to put statistical symbols besides each data point. Therefore, we plotted the data in a bar graph as shown in the Supplementary Figure S1 to accommodate all of the statistical symbols at each data points. We have added statistical symbols to Fig. 2 A & B. However, as Fig. 3A is the dose response curve, we focused on the curve fitting and the statistical symbol was not included. 

Reviewer #1: 

1. In figure 4, 5, S2 and S3, the incubation times of ZnO-NP treated cells were difference. The authors should explain why the difference.

Response: We are sorry for the confusion caused by the typos. The total time for ZnO-NP treatment in these experiments was 24 hr and stimulants were added at different time. For 6-OHDA treatment (Fig. 4 and 5), 6-OHDA was added at the 18th hr and the cells was used for MTT and ROS assays at 24th hr (Fig. 4B). For H2O2 treatment, H2O2 was added at the 20th hr for the viability assay (Fig. S4A) and 23rd hr for the ROS production assay (Fig. S4B). We have modified the manuscript to clarify the experiment (Page 6, Lines 129-131; Page 17, Lines 354-357; Legends of Figures S3 & S4). 

2. The authors need to discuss the possible reason or mechanism for explaining why ZnO-NP decreased ROS production and p53 expression.

Response: Our results here suggest that a weak-to-mild elevation of [Zn2+]i decreases the generation of ROS and protecting cells from 6-OHDA-induced death. It has been suggested that the elevation of [Zn2+]i can enhance the expression of metallothioneins and the activities of Zn2+-containing superoxide dismutases. Metallothioneins can work as antioxidants and dismutase can reduce the ROS level resulting in the downregulation of apoptosis pathway as indicated by the expression of p53 and bax/bcl-2 ratio. Therefore, ZnO-NP at a low dosage can provide a protection against the ROS-induced damages and cell death. (Page 14, Lines 299-304) 

3. Please indicate the molecular size of ZnT1 and ZIP8 in figure 2. The exact migration position of at least one independent molecular weight marker protein should be indicated with each gel or blot. Each blot panel should retain space above and below the band of interest from the original image.

Response: Thanks for the suggestions. Because of the space limitation, it is difficult to have all of the information in a figure. The original images containing the markers of the Western blots and agarose gels shown in Fig. 2, 3, and 4 were now included in the Supporting Information Figure S3, S4, and S7, respectively. 

4. Some spelling errors should be avoided. Such as, in discussion 9th line, “ snRNA” change “shRNA”.

Response: We are sorry for those typos and have checked the spelling of the manuscript carefully, like “German” to “Germany” (Page 5, Line 99), duplicated abbreviation in the abstract (Zinc oxide-nanoparticle (ZnO-NP), now deleted), changed “lose” to “low” (P12, Line 250) etc. 

Reviewer #2: Authors investigated the effects of Zinc oxide nanoparticles, which had little cytotoxicity, in cultured human neuroblastoma SH-SY5Y cells and characterized the importance of Zn2+ transporters in 6-hydroxy dopamine (6-OHDA)-induced cell death. The manuscript is well-written, with a significant summary and discussion.

Response: We thank the reviewer’s comments about the effect of ZnO-NP in protecting the cells from 6-OHDA-induced death.

---

## [Editor Report · Decision Letter 1]

28 Aug 2020

Zinc oxide nanoparticles modulate the gene expression of ZnT1 and ZIP8 to manipulate zinc homeostasis and stress-induced cytotoxicity in human neuroblastoma SH-SY5Y cells

PONE-D-20-11050R1

Dear Dr. Liu,

We’re pleased to inform you that your manuscript has been judged scientifically suitable for publication and will be formally accepted for publication once it meets all outstanding technical requirements.

Kind regards,

Academic Editor

PLOS ONE
---

## [Editor Report · Acceptance letter]

2 Sep 2020

PONE-D-20-11050R1 

Zinc oxide nanoparticles modulate the gene expression of ZnT_1_ and ZIP_8_ to manipulate zinc homeostasis and stress-induced cytotoxicity in human neuroblastoma SH-SY5Y cells  

Dear Dr. Liu:

I'm pleased to inform you that your manuscript has been deemed suitable for publication in PLOS ONE. Congratulations! Your manuscript is now with our production department. 

Kind regards, 

on behalf of

Dr. Hsi-Lung Hsieh 

Academic Editor

PLOS ONE